# Diagnostic Findings of Transmissible Viral Proventriculitis Associated with Chicken Proventricular Necrosis Virus in Processed Broiler Chickens in Argentina

**DOI:** 10.3390/v17040519

**Published:** 2025-04-01

**Authors:** Carlos Daniel Gornatti-Churria, Natàlia Majó, Melissa Macías-Rioseco, Rosa M. Valle, Patricio A. García, Carmen F. Jerry

**Affiliations:** 1Unidad Académica de Avicultura, Facultad de Veterinaria, Universidad de la República, Montevideo 13000, Uruguay; 2Institut de Recerca i Tecnologia Agroalimentàries (IRTA), Centre de Recerca en Sanitat Animal (CReSA), Universitat Autònoma de Barcelona (UAB), 08123 Barcelona, Catalunya, Spain; natalia.majo@irta.cat (N.M.); rosa.valle@irta.cat (R.M.V.); 3Departament de Sanitat i Anatomia Animals, Facultat de Veterinària, Universitat Autònoma de Barcelona (UAB), 08123 Barcelona, Catalunya, Spain; 4California Animal Health and Food Safety Laboratory System (CAHFS) Tulare Branch, School of Veterinary Medicine, University of California, Davis, CA 93274, USA; mmaciasrioseco@ucdavis.edu; 5Cátedra de Patología Aviar y Pilíferos, Cátedra de Zootecnia Especial III y Cátedra de Tecnología de los Alimentos, Facultad de Bromatología, Universidad Nacional de Entre Ríos, Gualeguaychú 2820, Argentina; patricio.garcia@uner.edu.ar; 6Bonnin Hnos. S.H., Colón 3280, Argentina; 7California Animal Health and Food Safety Laboratory System (CAHFS) Turlock Branch, School of Veterinary Medicine, University of California, Davis, CA 95380, USA; cfjerry@ucdavis.edu

**Keywords:** broiler chicken carcasses, chicken proventricular necrosis virus (CPNV), gastric isthmus, processing plant, proventriculus enlargement, RT-PCR, transmissible viral proventriculitis (TVP)

## Abstract

Transmissible viral proventriculitis (TVP) and chicken proventricular necrosis virus (CPNV) affect the broiler industry globally and are emerging diseases of economic importance. Here, we present the findings of TVP from processed broiler carcasses in Argentina following marked condemnation at the processing plant. We studied a total of 122 abnormally presenting proventriculi at processing from 42-to-50-day-old, male Cobb500^™^ broiler chicken carcasses from 11 farms belonging to the same company in 13 episodes of proventriculi–gizzards condemnation between December 2021 and April 2022. The proventriculi were enlarged and pale with a widened gastric isthmus. A histopathologic lesion score system was developed based on the presence of a combination of key microscopic findings, the distribution, and the severity of the lesions. Scoring of the affected proventriculi revealed 65% (79/122) with a score of 4, 23% (28/122) with a score of 3, and 12% (15/122) with a score of 2. Focal to multifocal immunoreactivity against the VP2-CPNV antigen within the necrotic glandular epithelial cells was noted in the affected proventriculi using immunohistochemistry. We found 84.4% (103/122) of the studied proventriculi with TVP lesions grossly and microscopically scored were positive for CPNV by RT-PCR. The sequencing results of the PCR product showed a high nucleotide sequence similarity (88.97%) to previously published VP1-CPNV sequences. We confirmed CPNV infection in most of the TVP affected proventriculi in all condemnation episodes at a broiler chicken processing plant in Argentina during the studied period. This study documents TVP associated with CPNV detection at processing plants in Argentina for the first time.

## 1. Introduction

Transmissible viral proventriculitis (TVP) is associated with infection by a *Birnavirus* closely related to the infectious bursal disease virus, currently named chicken proven-tricular necrosis virus (CPNV), which affects the broiler chicken industry worldwide [1,2]. CPNV was firstly identified causing TVP based on viral isolation from the proventriculi of affected chickens in 2005 in the USA [2]. Since then, CPNV infection associated with TVP has been reported in the field, mostly affecting commercially raised chickens in Alge-ria, Brazil, Catalonia, China, France, Hungary, Iraq, Poland, Republic of Korea, the UK, and the USA [3,4,5,6,7,8,9,10,11,12,13,14,15,16,17]. CPNV is most likely to be transmitted horizontally through the oral−fecal route. TVP can impair growth performance due to reduced feed digestion and poor feed conversion in the field. The presence of non-digested or poorly digested feed is noted in the feces. Grossly, the affected commercial chickens showed enlarged, pale proventriculi with thickened walls and a widened, friable, and weakened gastric isthmus [1,2]. At processing plants, gastric isthmus and proventriculi are prone to rupture, leading to the contamination and reprocessing of carcasses, and increased condemnation rates of viscera [1,2]. Proventriculus enlargement can be attributed to a wide range of infectious and non-infectious entities that affect the global commercial chicken industry; thus, TVP and CPNV infection need to be differentiated [1,2]. The aim of this study was to characterize the gross, microscopic, immunohistochemical, RT-PCR, and sequencing diagnostic findings of CPNV infection in condemned proventriculi at a broiler chicken slaughterhouse and processing plant in Argentina.

## 2. Materials and Methods

### 2.1. Case Study

A total of 20,000 severely enlarged, pale proventriculi with widened gastric isthmus (Figure 1A,B) were noted during the evisceration of 42-to-50-day-old, male Cobb500^™^ broiler chicken carcasses at a commercial slaughterhouse and processing plant in Entre Ríos Province, Argentina. In order to investigate these condemnation events, we studied a total of 13 episodes of proventriculi and gizzard condemnation affecting 11 farms belonging to one company from December 2021 to April 2022 (Appendix A).

### 2.2. Histopathology

A total of 122 proventriculi were sampled for histopathologic evaluation. Proventricular tissue sections were collected and fixed through immersion in 10% neutral-buffered formalin (pH 7.2) for 24–72 h and processed by standard histologic techniques to produce 4 μm, H&E-stained sections. The prolonged fixation of tissues in 10% neutral-buffered formalin occurred in few instances due to logistic problems of transport.

Histopathologic sections of each of the condemned proventriculi were evaluated for any microscopic lesions of TVP and then scored. These included lymphocytic follicular aggregates (LFAs), necrosis of glandular epithelium (NGE), hyperplasia/metaplasia of glandular epithelium (H/MGE), distension of proventricular glands (DPGs), and intraglandular lymphocytic inflammatory infiltration (ILII), together with the evaluation of their distribution (focal, multifocal, and diffuse) and severity (mild, moderate, and severe) being scored in increased severity from 0 to 4 (Appendix A).

### 2.3. Immunohistochemistry

Immunohistochemical staining was performed on a set of 15 proventriculi with the most severe microscopic lesions. A primary rabbit anti-VP2 CPNV polyclonal antibody was used according to standard methodologies of the California Animal Health and Food Safety Laboratory System (CAHFS), University of California–Davis [18].

### 2.4. RT-PCR and Sequencing Analysis

We performed RNA extraction and RT-PCR for CPNV detection on all 122 formalin-fixed, paraffin-embedded condemned proventriculi included in a total of 57 blocks, and the RT-PCR product from a positive case was selected and sequenced, following previously described methodology [15,16]. We studied the sampled proventriculi with the permission of sanitary authorities from the slaughterhouse, and followed standard procedures of the Argentinian commercial processing plants.

## 3. Results

### 3.1. Histopathology

A total of 112 proventriculi (92%) cases had microscopic lesions suggestive of TVP. Ten proventriculi (8%) showed no degenerative and necrotic glandular lesions; only multifocal interstitial lymphocytic inflammatory cell infiltration was noted. Based on the histopathologic scoring system of this study, we observed, in decreasing order, 65% (79/122) proventriculi with a score of 4 (Figure 2), 23% (28/122) with a score of 3, and 12% (15/122) with a score of 2 (Figure 3; Appendix A). We found no proventriculi with histopathologic scores of 0 and 1 in this study. We noticed other histolopathogic findings such as multifocal to coalescent cartilaginous metaplasia (1/122, 0.8%), granulomatous inflammation (1/122, 0.8%), and hemorrhages (2/122, 1.6%) within the proventricular glands.

### 3.2. Immunohistochemistry

We found focal to multifocal immunoreactivity within necrotic cells of the glandular epithelium in all 15 sections of proventriculi tested by immunohistochemistry (Figure 4).

### 3.3. RT-PCR and Sequencing Analysis

From the 122 collected and analyzed proventriculi with TVP-suggestive gross and microscopic findings, 84.4% (103/122) were confirmed positive for CPNV by RT-PCR, showing a high correlation between TVP-compatible findings and CPNV RT-PCR results. In decreasing order, proventriculi with histopathologic scores of 4 (50.8%, 62/122), 3 (22.1%, 27/122), and 2 (11.5%, 14/122) had RT-PCR CPNV positive results (Figure 5; Appendix A). Sequencing analysis from one of the RT-PCR positive cases had a high nucleotide sequence similarity (88.97%) compared to previously published VP1-CPNV sequences.

## 4. Discussion

To the best of our knowledge, this report describes the diagnosis of TVP associated with CPNV infection in commercially raised chickens in Argentina for the first time. We developed a histopathologic lesion score based on presence and combination of key microscopic lesions of TVP including LFA, NGE, H/MGE, DPG, and ILII and graded in extension and severity in condemned proventriculi with TVP gross lesions. We considered cartilaginous metaplasia as an uncommon microscopic lesion in a proventriculus, which had TVP-compatible gross and microscopic findings and a positive CPNV RT-PCR result. We postulate the lesion may have been triggered by chronic inflammation within the proventriculus. We associated the gross and histopathologic postmortem findings and RT-PCR results to detect CPNV in the majority of condemned proventriculi from all 13 episodes in a broiler chicken processing plant within the studied period. Here, we provide novel CPNV sequencing results that we compared with TVP−CPNV infection field cases uploaded in the GenBank database from Bangladesh, Hungary, Israel, Romania, Spain, the UK, and the USA. Previously reported microscopic scoring systems for TVP evaluating field cases of proventriculi enlargement fulfilled the TVP-diagnostic criteria in 54.9% and 48% of the evaluated proventricular samples in Brazil and the UK, respectively [4,16]. The use of microscopic scoring systems for proventriculi with TVP-compatible gross lesions in the field and processing plants could be a starting point for understanding how this disease progresses in chickens. We speculate that negative results for CPNV by RT-PCR in the remaining 15.6% proventriculi (19/122; 17 scored 4, 1 scored 3 and 1 scored 2) may be a true negative due to the lack of viral presence. But, they might also be due to the absence of a necrotizing process, predominant inflammatory responses, or the chronicity of lesions. False negatives may also potentially be attributed to prolonged fixation in 10% formalin, which can degrade viral RNA and compromise molecular detection of the etiologic agent. Although naturally occurring TVP and CPNV infections affecting broiler chicken farms have been described in Brazil, the USA, Europe, North Africa, the Middle East, and East Asia [1–17], we found no reports of TVP associated with CPNV infection at broiler chicken processing plants in a search of Google Scholar, PubMed, CAB Direct, Scopus, and Web of Science, suggesting there has been no such study previously reported in the current literature.

In our study, grossly affected and condemned proventriculi were still present at the broiler chicken processing plant we studied in Argentina after April 2022, but in the number and severity of lesions decreased [19]. but in the number and severity of lesions decreased [19], a similar pattern was seen in field reported findings in 2020 in California, the USA [17], and after implementing all-in-all-out husbandry practice, strict cleaning and disinfection procedures, and extended downtime.

Previous works related to TVP and CPNV infections on broiler chickens were based on case reports or retrospective studies in the field, without any correlation with proventriculi condemnation at the chicken processing plants [1–17]. Recent studies of the proventricular virome of broiler chickens affected by TVP under both natural and experimental conditions in China [5,20,21] and the Republic of Korea [14] revealed other viruses different from CPNV, including a novel cyclovirus [5], gyroviruses (gyrovirus homsa1, chicken anemia virus, and gyrovirus galga1) [20,21], and avian picornavirus [14]; however, we demonstrated the involvement of CPNV infection in most of TVP-compatible grossly and microscopically affected proventriculi by molecular methods and immunohistochemistry. Some other differentials for proventricular pathology in chickens include Marek’s disease, fowl adenovirus, *Macrorhabdus ornithogaster*, *Candida albicans*, *Dispharynx nasuta*, *Tetrameres* spp., and *Cryptosporidium* spp. infections [2]; non-infectious entities similarly producing proventricular enlargement in chickens, such as mycotoxins, high copper sulfate ingestion, and dietary biogenic amines, should also be considered [22].

In our study, veterinary inspectors and technicians played a key role in the identification of TVP-compatible grossly affected proventriculi during evisceration. Evisceration and washing carcasses is one of the risk points potentially related to the contamination and reprocessing of broiler chicken carcasses.

## 5. Conclusions

TVP associated with CPNV is an emerging disease, which, to the best of the authors’ knowledge, is described for the first time as causing proventricular condemnation at a broiler chicken slaughterhouse and processing plant in Argentina. Veterinary inspectors and technicians at the processing plant and the scoring of microscopic lesions both played a crucial role in the initial and presumptive anatomopathologic diagnosis of TVP. Altogether, light microscopy, RT-PCR, and sequencing ancillary tests provided the characterization of CPNV infection on condemned proventriculi in our study. We found a strong correlation between the scoring of microscopic lesions and RT-PCR CPNV-positive results. Control and preventive field measures in TVP−CPNV-affected flocks can mitigate the negative economic impact at the field, especially when no commercial or autogenous vaccines are available.

## Figures and Tables

**Figure 1 viruses-17-00519-f001:**
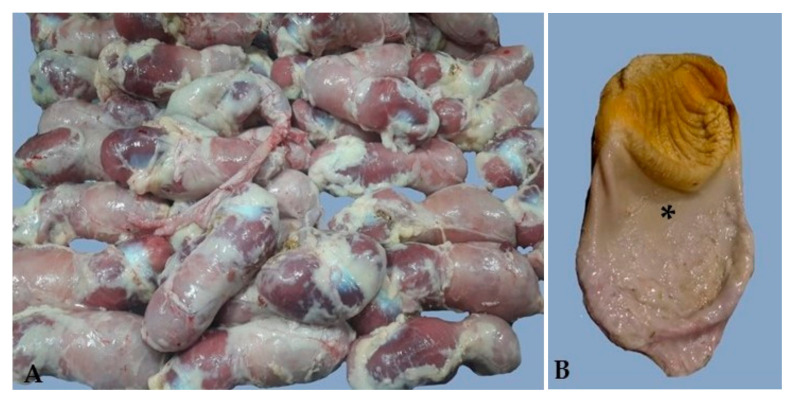
Proventriculi and ventriculi from 42-day-old, male Cobb500^™^ broiler chicken carcasses. (**A**) Numerous, severely enlarged and pale proventriculi with widened gastric isthmus from one of the condemnation episodes at a commercial broiler chicken processing plant in Argentina. (**B**) Opened proventriculus from the condemnation episode of Subfigure (**A**) showing an enlarged size and marked gastric isthmus (asterisk).

**Figure 2 viruses-17-00519-f002:**
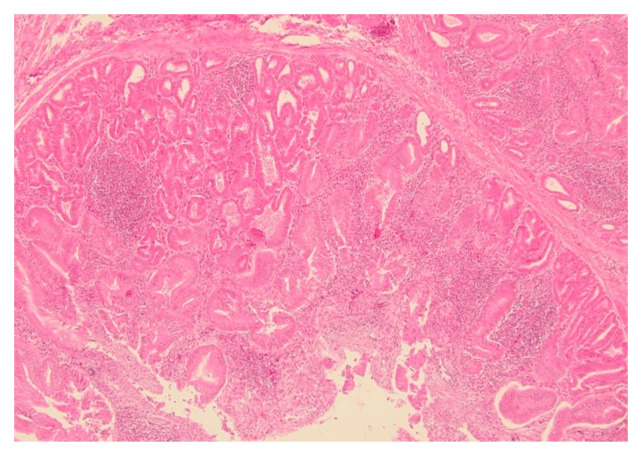
Proventriculus from a 50-day-old, male Cobb500^™^ broiler chicken carcass. Severe multifocal interstitial lymphocytic inflammatory cell infiltration of the submucosa, and glandular hyperplasia/metaplasia. H&E, objective 10×.

**Figure 3 viruses-17-00519-f003:**
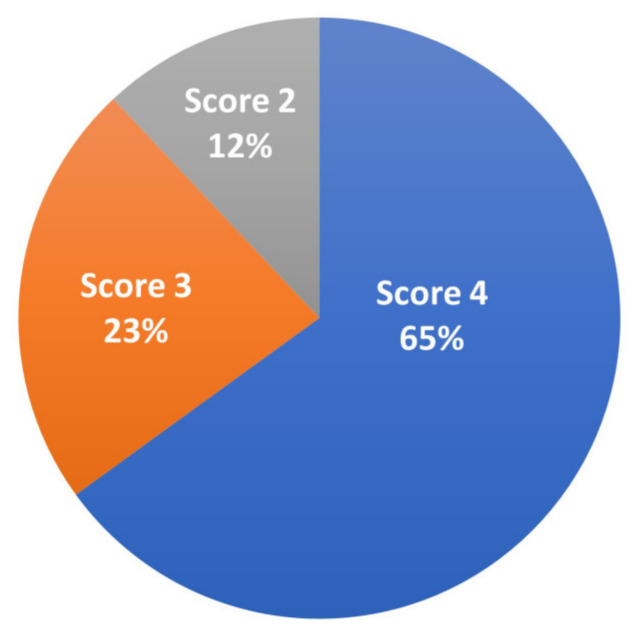
Percentage grossly affected proventriculi scored based on compatible microscopic lesions of CPNV infection diagnosed in 42-to-50-day-old male Cobb500^™^ broiler chicken carcasses at a processing plant in Argentina.

**Figure 4 viruses-17-00519-f004:**
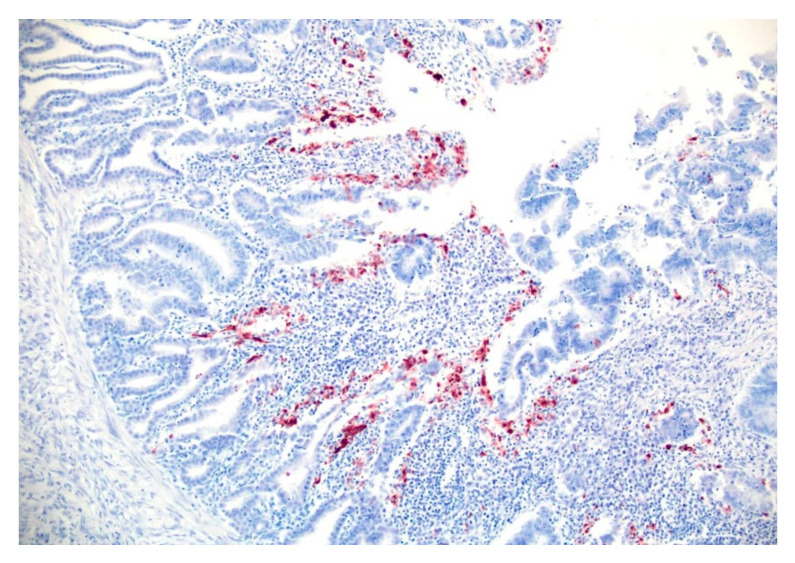
Proventriculus from a 47-day-old, male Cobb500^™^ broiler chicken carcass. Diffuse immunoreactivity within numerous necrotic epithelial glandular cells. Immunohistochemistry, diaminobenzidine counterstain. Objective 20×.

**Figure 5 viruses-17-00519-f005:**
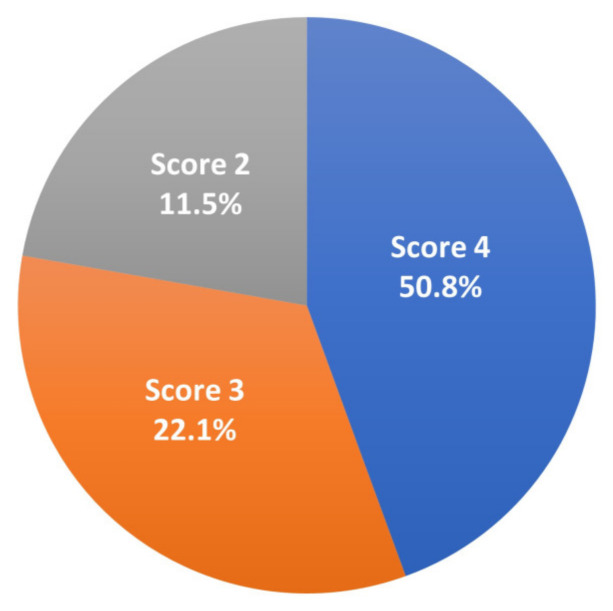
Scored proventriculi with microscopic lesions of CPNV infection and positive results for CPNV RT-PCR diagnosed in 42-to-50-day-old male Cobb500^™^ broiler chicken carcasses at a processing plant in Argentina. Note the 15.6% (19/122) scored proventriculi had a negative result for CPNV RT-PCR.

## Data Availability

The original contributions presented in this study are included in the article. Further inquiries can be directed to the corresponding author.

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
