# Peer review of "Diagnostic Findings of Transmissible Viral Proventriculitis Associated with Chicken Proventricular Necrosis Virus in Processed Broiler Chickens in Argentina"

_viruses, 2025, doi:10.3390/v17040519_

Round 1
Reviewer 1 Report
Comments and Suggestions for Authors
The reviewed paper presents very interesting data on TVP and CPNV in chicken broiler flocks in Argentina. Although the experiment was planned in an understandable and interesting way, unfortunately, the presentation of the layout and the results of the work means that the publication must undergo major revision.
The main points I would like to draw attention to:
- MATERIALS AND METHODS - this chapter is presented very fragmentary. The complete lack of detailed description of laboratory methods, makes it difficult to imagine the authors' reasoning.
- RESULTS - similar to the chapter „material and methods”, the results are presented in a sloppy and chaotic manner, which makes it hard to navigate through the material and thus draw conclusions.
Minor sugestions:
- Use „TVP” when describing HP lesions typical for a disease instead of „CPNV infection”
- There are discussion inserts in the results section
- Last paragraph of the Discussion is not the topic of the reasearch at all – I sugest to delete this part.
Author Response
REVIEWER 1
Thank you for your valuable review and opinions. Please find below the editing we made to answer your suggestions and changes
1.- 1. MATERIALS AND METHODS - this chapter is presented very fragmentary. The complete lack of detailed description of laboratory methods, makes it difficult to imagine the authors' reasoning. 2. RESULTS - similar to the chapter „material and methods”, the results are presented in a sloppy and chaotic manner, which makes it hard to navigate through the material and thus draw conclusions.
-. The subheadings "Case study", "Histopathology", "Immunohistochemistry" and RT-PCR and sequencing analysis" in Materials and Methods, and the subheadings "Histopathology", "Immunohistochemistry" and "RT-PCR and sequencing analysis" in Results sections were added to clarify the presentation of data in each of these two sections.
2.- Use „TVP” when describing HP lesions typical for a disease instead of „CPNV infection”.
-. The term "TVP" was used instead of "CPNV infection" describing histopathologic lesions in L82 and L158 in the new version of the manuscript. In L221, "CPNV infection" was deleted in the new version of the manuscript.
3-. There are discussion inserts in the results section.
-. The paragraphs L106-L109 ("We considered...by chronicity of proventricular lesions"), L117-L119 ("We associated the gross...within the studied period") and L121-L123 (Hence,... and the USA) from the previous version were all moved from the Results to the Discussion section, as suggested.
4.- Last paragraph of the Discussion is not the topic of the reasearch at all – I sugest to delete this part.
-. The paragraph describing Salmonella and Campylobacter contaminations of carcasses was deleted, as suggested.
Reviewer 2 Report
Comments and Suggestions for Authors
Diagnostic findings of TVP associated with Chicken Proventricular Necrosis Virus in Processed Broiler Chickens In Argentina by Gornatti-Churra et al is a manuscript that documents TVP and CPNV in proventriculi that are condemned at processing.
The authors did a good job describing what they did. There are just a couple of items that need to be addressed or clarified.
Line 71. belonged should be belonging.
Line 215. replace histopathologic with light microscopy
Line 244. was employed by are employed by....needs clarification. Does this person still work for the company or did they and not any more.
Figures.
Figure 1. I think adding an arrow on the 1B photo that shows the gastric isthmus would be helpful with readers who are not quite as knowledgable about poultry organs.
Figure 2. Is this really taken at 10X or is that the slide objective only? The true magnification is the eye objective (typically 10X) times the slide objective. so 10X x 10X equals 100X.
Figure 4. Same comment as figure 2. 20X or 200X?
Author Response
REVIEWER 2
Thank you your valuable opinion on this review. Please find below our editing following your suggestions and changes you proposed.
1-. Line 71. belonged should be belonging.
The edit was made as suggested (L73 of the revised manuscript).
2 -. Line 215. replace histopathologic with light microscopy
"Light microscopy" was used to replace "histopathology" and "immunohistochemistry", as suggested (L222 of the revised manuscript).
3 -. Line 244. was employed by are employed by....needs clarification. Does this person still work for the company or did they and not any more.
This is an statement previously used as suggested from other MDPI journal in which two of the authors (CDGCh and PAG) published other results obtained from the studied processing plant. The Managing Editor of this manuscript could suggest editing on it, if necessary.
Figures.
4 -. Figure 1. I think adding an arrow on the 1B photo that shows the gastric isthmus would be helpful with readers who are not quite as knowledgable about poultry organs.
An asterisk was added to the Figure 1B to point out the widened gastric isthmus and figure caption (L133), as suggested.
5 -. Figure 2. Is this really taken at 10X or is that the slide objective only? The true magnification is the eye objective (typically 10X) times the slide objective. so 10X x 10X equals 100X.
Editing was made as suggested (L137 of the revised manuscript).
6 -. Figure 4. Same comment as figure 2. 20X or 200X?
Editing was made as suggested (L147 of the revised manuscript)
Round 2
Reviewer 1 Report
Comments and Suggestions for Authors
Minor comments:
line 74 - correct the sentence
Line 80 and 159 - histopathologic, not histologic
Line 89 - severe, not severely
Line 271 onward - correct the sentence
Line 303 onward - correct the sentence
Comments on the Quality of English LanguageIn this reviewers opinion the English could be improved.
Author Response
Thank you for your valuable review. Please find below the edits addressing your suggestions and changes, together with the english language editing of the manuscript, as suggested.
-. Line 74 - correct the sentence
The sentence was edited as suggested (L75).
-. Line 80 and 159 - histopathologic, not histologic
Both edits were included, as suggested (L81 and 109).
-. Line 89 - severe, not severely
The edit was included, as suggested (L90).
-. Line 271 onward - correct the sentence
This sentence ("and how effective ... are in affected farms") was deleted.
-. Line 303 onward - correct the sentence
This paragraph ("Continuous education ... of the chicken meat market") was deleted.